# *Legionella pneumophila*: The Journey from the Environment to the Blood

**DOI:** 10.3390/jcm11206126

**Published:** 2022-10-18

**Authors:** Valeria Iliadi, Jeni Staykova, Sergios Iliadis, Ina Konstantinidou, Polina Sivykh, Gioulia Romanidou, Daniil F. Vardikov, Dimitrios Cassimos, Theocharis G. Konstantinidis

**Affiliations:** 1Izhevsk State Medical Academy, Kommunarov Street 281, 426034 Izhevsk, Russia; 2Faculty of Public Health, Medical University of Sofia, Byalo More Str. 8, 1527 Sofia, Bulgaria; 3Medical School, University of Patras, Rio, 26504 Patras, Greece; 4State Budgetary Health City Polyclinic No 2 (GBUZ GB2) of Krasnodar, Seleznev Street 4/10, 350059 Krasnodar, Russia; 5Nephrology Department, General Hospital “Sismanogleio”, 69100 Komotini, Greece; 6Russian Research Center for Radiology and Surgical Technologies of the Ministry of Health of the Russian Federation, Tkachey Str. 70-16, 192029 St. Petersburg, Russia; 7Pediatric Department, Democritus University of Thrace, 68100 Alexandroupolis, Greece; 8Blood Transfusion Center, University General Hospital of Alexandroupolis Dragana Campus, 68100 Alexandroupolis, Greece

**Keywords:** *Legionella pneumophila*, bloodstream infection, bacteremia

## Abstract

An outbreak of a potentially fatal form of pneumonia in 1976 and in the annual convention of the American Legion was the first time that *Legionella* spp. was identified. Thereafter, the term Legionnaires’ disease (LD) was established. The infection in humans is transmitted by the inhalation of aerosols that contain the microorganisms that belong to the Legionellaceae family and the genus Legionella. The genus Legionella contains genetically heterogeneous species and serogroups. The *Legionella pneumophila* serogroup 1 (Lp1) is the most often detected strain in outbreaks of LD. The pathogenesis of LD infection initiates with the attachment of the bacterial cells to the host cells, and subsequent intracellular replication. Following invasion, *Legionella* spp. activates its virulence mechanisms: generation of specific compartments of Legionella-containing vacuole (LCV), and expression of genes that encode a type IV secretion system (T4SS) for the translocation of proteins. The ability of *L. pneumophila* to transmigrate across the lung’s epithelium barrier leads to bacteremia, spread, and invasion of many organs with subsequent manifestations, complications, and septic shock. The clinical manifestations of LD depend on the bacterial load in the aerosol, the virulence factors, and the immune status of the patient. The infection has two distinct forms: the non- pneumatic form or Pontiac fever, which is a milder febrile flu-like illness, and LD, a more severe form, which includes pneumonia. In addition, the extrapulmonary involvement of LD can include heart, brain, abdomen, and joints.

## 1. Introduction

More than 46 years ago, during a meeting of the American Legion in Philadelphia, USA, an outbreak of pneumonia appeared [1]. The *Legionella* spp. was originally isolated and described in 1947 as a “rickettsia-like” organism, and only in 1977 was this organism retrospectively identified as the same species and serogroup as the Philadelphia bacterium [2]. Overall, 182 cases were reported, and 29 of them were fatal. Although the bacterium strain accounted for this outbreak was not found, it was postulated that the microbial strains belonged to an air borne microorganism [3] and the potentially fatal pneumonia was termed Legionnaires’ disease (LD) [4]. The etiological factor of Legionnaires’ disease was identified as *Legionella pneumophila* (*L. pneumophila*) [5]. After its identification, *L. pneumophila* was classified as a ubiquitous environmental bacterium that belonged to the Legionellaceae family. It is a small, Gram-negative, aerobic, non-sporogenous, and non-capsule-forming bacilli that possesses the enzymes catalase and oxidase. The guanine and cytosine content in its DNA ranges from 38% to 52% [6]. In addition, *L. pneumophila* is a mobile bacterium. Interestingly, it was shown that the mobility of *L. pneumophila* differs between serogroups [7]. One of the most interesting characteristics of legionellae is thermotolerance. *L. pneumophila* is viable between 25 and 37 °C, but there is no significant in its multiplication at 46 °C [8]. *L. pneumophila*, as a water-living bacteria, tends to adhere to surfaces and develops an organic protective matrix, creating microenvironments known as biofilms. The ability to form biofilms allows microbes to survive for long periods. Biofilms play an important role in the survival of microorganisms in low-nutrient environments. Marin et al. reported that *Legionella pneumophila* regulates biofilm formation through the bffA gene [9]. Moreover, biofilm is a form of microbial community that enables the transfer of genetic material to differentiate bacterial virulence factors [10].

Infections in humans arise exclusively by the inhalation of aerosols that contain microorganisms, which can occur in air conditioning systems, cooling towers, spas, fountains, ice machines, plant sprayers, dental appliances, and showerheads. Moreover, person-to-person transmission of Legionnaires’ Disease (LD) was also reported [11].

The aim of this review is to analyze the epidemiology of infections due to *Legionella* spp. and relate the basic biology and virulence factors of *Legionella* spp. to its pathogenesis at the cellular level, as well as analyze the clinical manifestation of LD.

## 2. Epidemiology

According to the annual report of the European Centre for Disease Prevention and Control (ECDC), case rates of Legionnaire’s disease in 2019 in the EU/EEA were of 2.2 cases per 100,000 individuals, with the highest rate reported in Slovenia (9.4 cases per 100,000 individuals) [12]. Legionella strains are widespread in water sources and in human-made environments [13]. Potential sources of *Legionella* spp. transmission are potable water sources, such as showers and taps, and non-potable sources, such as fountains, spas, cooling towers, and evaporative condensers [14]. Global climate change favored the amplification of Legionella colonization in aging water supply and water infrastructure of both urban and suburban areas. Moreover, Alexandropoulou et al. reported that car cabin air filters represent a reservoir of Legionella bacteria, and thus a potential pathway for contamination [15]. Velonakis et al., who isolated the Legionella species including the *L. pneumophila* serogroup 1 in Greek potting soils, made the hypothesis that potting soils may constitute a reservoir for legionella strains [16]. In addition, Oda et al. reported Legionella pneumonia following the heavy rain event of July 2018 in Japan, in a 62-year-old man who lives in a flooded area [17]. In addition to all of the above, infections can also be associated with recreational and garden areas of hotels [18].

*Legionella pneumophila* is a genetically heterogeneous species, composed of a total of 32 Legionella species and 51 serogroups [19,20]. The *Legionella pneumophila* serogroup 1 (Lp1) is one of the most commonly detected strains that causes outbreaks worldwide [21]. It was also demonstrated that *L. pneumophila* consists of three subspecies: pneumophila, fraseri, and pascullei [22,23]. Furthermore, LD can be caused by non-*L. pneumophila* species, such as *Legionella longbeachae* [24].

The prevalence of LD and Pontiac fever (PF) has increased in recent decades. The disease develops as sporadic cases or as outbreaks. LD is usually classified as community-acquired (CALD), travel-associated (TALD), or healthcare-associated (HALD). Travel-associated LD, as well as ship-associated events, occur repeatedly. Hotel-associated cases are usually linked to hotel cooling towers and/or potable water systems. Ship-associated cases are most commonly linked to hot tubs [25]. LD is rare in children and infants [26]. Only a small number of cases of *Legionella* spp. presented in neonates are described in the literature [27,28].

Cases of LD in occupational exposure are widely reported. Previous studies show that healthcare workers, hotel staff, professional divers, both car and bus drivers, as well as dental workers are among the professions that belong to high risk groups [29,30,31,32]. Additionally, upon testing healthy subjects who belonged to high risk groups, it was reported that they showed a higher titer of *L. pneumophila* antibodies [33,34]. Sawano et al. have presented a fatal case of LD, which occurred in a decontamination worker after the Fukushima nuclear disaster [35].

### The Impact of the Coronavirus Disease (COVID-19) Pandemic on Legionella *spp.* Epidemic

The severe acute respiratory syndrome coronavirus 2 (SARS CoV-2) disease is a pandemic that was firstly reported in Wuhan, China, in December 2019 [36]. Due to the rapid spread of the infection worldwide, the pandemic affected our everyday life and had a negative impact on healthcare services [37,38]. Prolonged building closures during the COVID-19 pandemic led to a stagnation in building water systems. As a result, the risk of colonization by *Legionella* spp. strains increased. The clinical manifestation of severe COVID-19 is similar to LD in some of the infected patients, with both pulmonary and extrapulmonary manifestations. While one’s clinical form can mimic the other, the coexistence of both diseases has also been described [39,40]. On the other hand, the pandemic has impacted airborne/droplet-transmitted diseases. Fischer et al. explored the impact of the COVID-19 pandemic on the reported case numbers of legionellosis in Switzerland. According to this article, the annual rate for legionellosis cases increased from 1.1/100,000 individuals in 2000 to 5.6/100,000 in 2020. Interestingly, the highest notification rate was recorded in 2018, during the pre-COVID-19 period, with 6.7/100,000 individuals [41].Tang et al. published an article with the same results. In this study, the authors compared the number of cases of airborne/droplet-transmitted notifiable infectious disease (NID) between the pandemic period and the pre-pandemic period. The occurrence of most diseases, including legionellosis in Taiwan, was lower during the pandemic period in comparison to the pre-pandemic period [42]. In contrast to most published data, Chao et al. reported that cases of infection due to *Legionella* spp. increased during the COVID-19 pandemic. At the same time, other infections of the lower respiratory tract such as influenza, invasive pneumococcal disease, or tuberculosis have been reduced [43].

## 3. The Mechanism of LD

In order to understand the disease’s mechanism, it is essential to acknowledge the virulence factors of *Legionella* spp. The pathogenesis of LD initiates with the attachment of the bacterial cells to the host cells, and includes intracellular replication and spreading. The ability of *L. pneumophila* to transmigrate across the barrier of the lung epithelium leads to bacteremia, septic shock, organ infiltration, relevant symptoms, complications, and finally multi-organ failure.

Following the entry of *L. pneumophila* into the host cells, it affects their function, creating a protective environment that provides nutrition for its growth [44]. The mechanisms involved include: apoptosis, autophagy, mitochondrial dynamics, and phospholipid biosynthesis [45,46]. *Legionella* spp. influences numerous eukaryotic cellular processes. This microorganism is characterized by its ability to translocate more than 300 effector proteins into the host’s macrophage. It was also shown that *L. pneumophila* inhibits eukaryotic protein synthesis by targeting mRNA translation [47]. Through genome sequencing, it was discovered that *L. pneumophila* encodes a variety of proteins similar in functionality to eukaryotic proteins [48]. Herkt et al. studied microRNAs (miRNAs) as critical posttranscriptional regulators of gene expression in bacterial infections. In this study, a trio of miRNAs (miR-125b, miR-221, and miR-579) was found to significantly affect intracellular *L. pneumophila* replication in a cooperative manner. Using proteome analysis, they found a downregulation of galectin-8 (LGALS8), DExD/H-box helicase 58 (DDX58), tumor protein P53 (TP53), and then MX dynamin-like GTPase 1 (MX1) by these three miRNAs [49].

### 3.1. Virulence Factors

*L. pneumophila’s* virulence factors, which are involved in different phases of the pathogenesis, have been described in the literature [50,51,52,53]. Some of these virulence factors are present in pathogenic bacteria, while they are absent in nonpathogenic strains.

#### 3.1.1. Structure of Bacterial Cells as Source of Surface Virulence Factors

The interactions of *Legionella* spp. with host organisms depends to a great extent on their surface structures, such as the cell envelope. Therefore, the ability of this intracellular bacterium to cause disease predominantly depends on the components and characteristics of its cell envelope. Initially, the bacterial cytoplasm is circumscribed by the inner membrane (Figure 1). It consists of bilayer phospholipids with integral proteins such as FeoB or metabolic enzymes, like IraA/IraB. The lipid structure and physicochemical properties of the inner membrane of *L. pneumophila* were discovered and analyzed by Hindahl and Iglewski [54]. It predominantly contains phosphatidylethanolamine and phosphatidylcholine. On account of the influential role of iron during all phases of *Legionella* spp. growth, there are different proteins and enzymes that become involved in this process. The uptake of Fe(II) is mainly carried out by FeoB, a GTP-dependent iron transporter. Moreover, in vitro studies have shown that FeoB is required for the efficient inactivation of macrophages [55]. Except from FeoB, the iron uptake mechanism also involves the proteins IraA and IraB, methyltransferases that mediate iron uptake and are required for the growth of human macrophages [54].

The periplasmic space, or periplasm, based on its property, contains soluble enzymes, such as proteases and nucleases, which has been considered an evolutionary precursor of the classical lysosomes of the eukaryotic cells [56]. Moreover, the periplasm contains one of the most important substances, a peptidoglycan, that plays a consequential role as a microbial virulence factor and contributes to the survival of *L. pneumophila* inside macrophages and amoeba. Furthermore, in this space, there are several co-localized antigens, including the peptidoglycan-associated lipoprotein (PAL) protein [53,57]. Interestingly, the PAL protein excretes in urine as a component of Legionella antigens and is therefore useful as an alternative diagnostic urinary antigen test.

The outer membrane (OM) is the distinguishing structural characteristic of *Legionella* spp. It is a lipid bilayer composed of phospholipids, lipoproteins, (LPS), and other proteins that connect the OM to the peptidoglycan (Figure 1). *Legionella* spp. contains several phospholipases on the OM which play a pathogenetic role as major virulence factors. Initially, bacterial phospholipases such as PlaB destroy host membranes and can manipulate the host’s signaling pathways [58]. Unfortunately, due to the hydrophobic nature of membrane proteins, and consequently their low solubility, the studies of OM proteins have been delayed. Studies revealed that the presence of surface-associated proteins on the OM, among the macrophage infectivity potentiator (Mip) protein and the major outer membrane protein (MOMP), play a critical role in legionellosis (Table 1).

The *L. pneumophila* major outer membrane protein (MOMP) is involved in the attachment to host cells. The MOMP was isolated by Hindahl and Iglewski [54]. Moreover, authors report that this protein is exposed at the cell surface. MOMP is encoded by the gene ompM [68]. Yang et al. reported that the MOMP may inhibit the chemotactic activity of the immune cells. In addition, the expression of several proteins related to the immune response such as interleukins -10, forkhead transcription factor 1 (FOXO1), nucleotide-binding oligomerization domain protein (NOD) 1, and other proteins was increased after exposure to MOMP [59].

The macrophage infectivity potentiator (Mip) protein is an essential virulence factor during the invasion process of *L. pneumophila*. Mip displays peptidyl–prolyl cis/trans isomerase activity and is necessary for the early stages of intracellular infection and survival in macrophages and pulmonary epithelium. The regulatory mechanism of the Mip protein during the process of bacterial infection of host cells is not yet completely understood. Shen et al. reported that Mip may influence macrophages on phagocytic and chemotactic activities [61]. It was shown, on the murine macrophage cell line (RAW264.7), that Mip increases the cells’ chemotaxis and inhibits the phagocytosis. Furthermore, by targeting *L. pneumophila ’s* macrophage infectivity potentiator (Mip) gene from environmental and clinical samples, a specific nested polymerase chain reaction for rapid detection was developed [69]. Interestingly, the targeting on the Mip gene has been recognized as the only useful tool for Legionella non-pneumophila species identification in samples [70]. Consequently, the Mip may help design vaccines, as it has already been proposed by He et al. [71].

Lipopolysaccharides (LPSs) are the main antigens of all Gram-negative bacteria, but the *Legionella* spp. LPSs cell-mediated responses are downregulated due to their carbohydrate nature [72]. Nevertheless, studies showed that LPSs are involved in the process of adhesion to the host cell [60].

Except from cells’ envelope proteins, there are numerous proteins that play an important role in legionellosis. The *L. pneumophila* PilY1 protein has a high preference towards other Gram-negative microorganisms such as *Pseudomonas aeruginosa*. This protein contains a C-terminal PilY domain and a domain homologous to a von Willebrand factor A (vWFa) [62]. The vWFa domain increases the ability of microbes to invade non-phagocytic cells.

The flagellum regulates *L. pneumophila*’s motility and has many other vital traits to their virulence. Rodgers et al. described the presence of flagella and pili structures on the *L. pneumophila* surface in 1980 [73]. Nevertheless, the mutant strains rpoN, fleQ, fliA, and flaA are non-flagellated and non-motile [74].

#### 3.1.2. Secreted Virulence Factors

*Legionella* spp. secretes various pigments, toxins, and enzymes, as well as more than 10,000 secretory proteins [75], some of which empower the *Legionella* spp. to survive in adverse environmental conditions, overcoming harmful factors such as light, and prevail against other microorganisms. During the stationary phase of its life cycle, when the bacteria have stopped replicating, *L. pneumophila* secretes a molecule called homogentisic acid (HGA), which is produced from the catabolism of amino acids’ tyrosine and phenylalanine. HGA combines with oxygen and forms a dark brown pigment called pyomelanin [76], which helps it acquire iron, an essential micronutrient. In addition, Levin et al. have shown that HGA also has toxic properties that can defend Legionella communities from invading microbes [77].

During the human infection state, *Legionella* spp. uses different secretion systems as source of virulence factors. There were four types of secretion systems described:

Type I secretion system (T1SS), known as Lss, consists of the ABC transporter (ATP-Binding Cassette), a membrane fusion protein, and an outer-membrane protein, which are exclusively identified in *L. pneumophila* strains [64]. T1SS is described in Gram-negative bacteria as a secretory system that secretes adhesins, iron-scavenger proteins, and lipases into the extracellular space [78].

Type II secretion system (T2SS), termed Lsp, secretes over 25 proteins and numerous degradation enzymes, including RNases and metalloproteases [65]. It was previously demonstrated that the T2SS substrate NttA promotes the intracellular growth of microbes. T2S promotes bacterial surveillance in lungs, both in macrophages and epithelial cells, and downregulates the host innate immune response [79]. The Lsp secretion system is essential for *L. pneumophila* survival at low temperatures.

Type III secretion system (T3SS) consists of a protein transport mechanism that translocates cytoplasmic substrates directly into the host’s cytoplasm. These secretion machines evolved from the bacterial flagella [66].

Type IV secretion system (T4SS) plays a critical role in the pathogenesis of *L. pneumophila* infection. The biological action of the type IV secretion system includes entrance into the host cells, replication, apoptosis, and finally egress from the host cells. There are two subclasses of the T4SS, namely the T4SS -A (Lvh) and the T4SS-B (Dot/Icm:(Defect in organelle trafficking/Intracellular multiplication). The Dot/Icm genes encode Dot/Icm secretory system proteins [67], and are vital for organelle trafficking and intracellular multiplication [80], while the Lvh secretion system contains genes encoding mobility factors and enzymes [81]. Furthermore, the Lvh can functionally replace defective Dot/Icm. T2SS and T4SS are found in all Legionella strains, whereas the T1SS is exclusive to *L. pneumophila*.

### 3.2. Macrophages

Legionella’s pathogenicity starts with microbial entry into the human organism with subsequent bacterial replication within phagocytes, such as alveolar macrophages. Macrophages play a critical role in the pathogenesis of LD (Figure 2). Earlier studies show that TLR3 and TLR4, along with TLR2 and TLR5, are major PRRs for *L. pneumophila*’s recognition by human macrophages [82,83]. The intracellular survival strategy of *Legionella* spp. is achieved by the orchestrate of endosomes, endoplasmic reticulum, Golgi apparatus, and mitochondria [44]. Following the invasion, *Legionella* spp. activates its virulence mechanisms: Legionella-containing vacuoles (LCVs), activation of the expression of genes that encode a type IV secretion system (T4SS) for the translocation of proteins [84], and secretion of small GTPase RAB1 to the LCVs’ membrane through the T4SS system, resulting in the redirection of vesicular transport between the ER and the Golgi system. *L. pneumophila* also ensures nutritional supply from mitochondria for bacterial replication [44].

In addition, *L. pneumophila* disposes autophagy inhibitory mechanisms via the effector protein, RavZ. This protein localizes onto the phagophore membrane (the precursor of the autophagosome) and then cleaves the C-terminal region of phosphatidylethanolamine (PE)-conjugated Atg8 family proteins such as LC3. The maturation of the LC3 protein is a cornerstone process in the phagophore elongation steps [85].

### 3.3. Lung Epithelium

The alveolar surface is lined up with the lung epithelium. Most of the surface area consists of type I pneumocyte cells that are specialized in gas exchange, while more sparse type II pneumocytes are responsible for secretory functions [51]. The ability of *Legionella* spp. strains to penetrate the alveolar epithelial barrier during the early phase of infection has a significant influence on the progress of Legionnaires’ disease. Wagner et al. reported that *Legionella* spp. transmigrate through a barrier of lung epithelial cells by the Legionella virulence factor Mip (macrophage infectivity potentiator) [86] that is binding to collagen. Moreover, the recombinant *Escherichia coli* strains (HB101) that contain *L. pneumophila*’s Mip protein can transmigrate across a barrier of lung epithelial cells.

### 3.4. Endothelial Cells

Previous studies have demonstrated that *L. pneumophila* may infect and breach the anatomic barrier of the lung—the pulmonary blood vessels—and endothelial cells as a potential initiating event in bacteremia, during the systemic spread of the bacteria via the blood’s circulation. Chiaraviglio et al. have shown that *L. pneumophila* can grow within cultured human endothelial cells [87].

### 3.5. Humoral Immune Response

Humoral immunity plays a secondary role in the host’s defense. In vitro and in vivo studies show that *L. pneumophila* induces dendritic cells’ (DCs) maturation [88,89]. The mechanisms of the DCs maturation depends on the signaling of Toll-like receptor 2 and 4 [90]. The antigen (Ag) presentation by DCs is evidenced by the up-regulation of MHC class II. The consequence of synthesis and the presence of specific antibodies fortify the phagocytosis. DCs activate antigen-specific CD4+ T cells, in an MHC class II-restricted manner [91]. Thereafter, activated CD4+ T cells stimulate antigen-specific B cells. Humoral immunity initially responds by producing IgM antibodies followed by IgG in order to activate the adaptive immunity against Legionella, and especially against reinfection. Moreover, matured DCs increase the secretion of proinflammatory cytokines such as IFN γ, TNF-α, IL-6, and IL-1β [92]. Furthermore, Neild et al. showed that dendritic cells (DCs) restrict the growth of intracellular Legionella [93]. In vitro studies have shown that *Legionella* spp. induces TNF, a secretion from macrophages [94].

## 4. Clinical Manifestation

Legionellosis is a generic term describing the pneumonic and non-pneumonic forms of human infection with *Legionella* spp. The clinical manifestation of LD depends on the bacterial load in the aerosol, the virulence factors, and the personal immune status of each patient. Moreover, risk factors include cigarette smoking and chronic lung disease [95]. *Legionella* spp. combined with other microorganisms, such as *Streptococcus pneumoniae*, *Helicobacter cinaedi* [96,97], and viral infections such as SARS CoV-2 [40], causes severe infection and sepsis. Interestingly, Sanchez et al. recently reported co-infections of *Legionella pneumophila* serogroup 1 with methicillin-resistant *Staphylococcus aureus* (MRSA) in a COVID-19 patient. The patient’s successful recovery was reported [39].

The infection has two distinct forms: the non-pneumonic form or Pontiac fever, which is a milder, febrile, flu-like illness, and Legionnaires’ disease, a more severe form of infection which includes pneumonia. Moreover, extrapulmonary manifestations of Legionnaires’ disease (LD) such as cardiac, brain, abdominal (gallbladder), joints, and skin involvement were reported [98]. In addition, manifestations of LD presenting with exanthem are extremely rare. There have been eleven cases reported in the literature of Legionellosis associated with rash [99].

Pontiac fever is an acute, self-limiting influenza-like illness that usually lasts 2–5 days [100]. The first documented outbreak of this form of legionellosis was reported in 1968 in Pontiac, Michigan, and affected at least 144 people [101]. The incubation period is usually up to 48 h and follows the onset of the disease symptoms. The main symptoms are fever, headache, malaise, and muscle pain (myalgia) [102].

The pneumonic form of legionellosis is Legionnaires’ disease (LD). The incubation period is usually 2 to 10 days, but it has been recorded in some outbreaks to be of up to 16 days. The severity of the disease ranges from a mild cough to a rapidly fatal pneumonia. Initial symptoms include fever, loss of appetite, headache, malaise, and lethargy. Some patients may also experience myalgia, diarrhea, and confusion. [102,103,104]. Usually, the presenting symptom is a mild, dry cough, that in 50% of the patients becomes productive. Hemoptysis occurs in about one-third of the cases. Some patients with immune suppression such as due to a kidney transplant or with SLE developed pulmonary abscess and pleural empyema [105,106]. Gastrointestinal manifestations such as watery diarrhea and sudden abdominal pain can be the presenting symptoms in patients with LD [107]. *L. pneumophila*, along with other intracellular pathogens, shares the propensity to produce relative bradycardia in the presence of fever. This clinical sign (Faget’s sign) is uncommon in bacterial pneumonia. Patel et al. described an even more rare manifestation of LD, namely hypertriglyceridemia and massive rhabdomyolysis, in a patient with disseminated legionella [104].

The death rate related to LD depends on the severity of the disease, the appropriateness of the initial antimicrobial treatment, the setting wherein legionella was acquired, and the host factors (nutrition, immune status, co-infection, etc.) [108]. Overall, the death rate is usually within the range of 5–10%. Nevertheless, it can reach up to 40–80% in untreated immuno-suppressed patients. With the appropriate management, the death rate, even in the above cases, can be reduced to 5–30% [109]. It is important to mention that healthcare-associated Legionnaires’ disease (HCA LD) can cause nosocomial outbreaks with high death rates [110].

Several laboratory abnormalities have been linked in the past to the diagnosis of legionellosis. These abnormalities include hyponatremia, hypophosphatemia, increased liver enzyme levels, and acute increase in creatine phosphokinase level [111]. Note that hyponatremia has been linked with severe legionellosis. Clinical studies have been inconclusive in identifying and verifying clinical and laboratory parameters that are specific for legionella infection. Individually, clinical and laboratory abnormalities lack diagnostic specificity [112,113]. A recent study by Beekman et al. validates a diagnostic scoring system for the detection of pneumonia due to *L. pneumophila*, based on six items on admission (Legionella prediction score) [114].

## 5. Bacteriemia

The Legionella species is an important cause of community- and hospital-acquired pneumonia. Bacteremia related to pneumonia by *L. pneumophila* is rarely reported. Lai et al. presented a hospital-acquired pneumonia and bacteremia caused by *L. pneumophila* in a patient with idiopathic thrombocytopenic purpura [115]. In addition, Nagase et al. reported bacteremia due to *L pneumophila* in a 74-year-old woman [97]. A rare manifestation of *Legionella* spp. in the lungs is diffuse alveolar hemorrhage [116]. *Legionella* spp. can cause bacteremia which can subsequently involve many organs and systems. As a result, the involvement of the heart can cause endocarditis and myocarditis. Legionella endocarditis is an infection with vague resemblance to bacterial endocarditis [117]. Acute myocarditis in patients with legionellosis can have clinical complications of multisystemic involvement (lung, heart, and kidney) [118]. Nishino reports another complication of bacteremia due to *Legionella* spp. which can be a mild encephalitis/encephalopathy with a reversible splenial lesion (MERS) [119]. The author described that the patient’s symptoms, as well as MRI imaging findings, improved on the 13th day of hospitalization, after treatment with intravenous levofloxacin, immunoglobulin, and methylprednisolone. In addition, legionellosis can often be associated with neurological symptoms, which is an indication of the hematogenic complication of *Legionella* spp. bacteremia [120,121]. Another rare complication of the *Legionella* spp. infection is immune thrombocytopenic purpura (ITP). The first case was reported in 1982 by Riggs et al. [122]. Recently, Javed et al. reported a case of a 61-year-old woman with critical bleeding and severe thrombocytopenia [123].

## 6. Diagnosis of Legionella Diseases

The gold standard for the diagnosis of an LP infection is the detection of specific microorganisms in clinical specimens obtained from patients, such as sputum and bronchoalveolar lavage (BAL), as well as serum or urine. Bacterial culture with the identification of microorganisms by serological and/or antibody-based assays, as well as polymerase chain reaction (PCR) are the main diagnostic methods. Urinary antigen tests (UATs) are widely used to diagnose Legionnaires’ disease. It is easy to perform a rapid assay without specialized skills and methods [124]. Unfortunately, the old urinary antigen tests were only able to detect the *L pneumophila* serogroup 1. The urinary antigen test (UAT) is highly accurate in order to diagnose infections due to the *L. pneumophila* serogroup 1. However, it is inadequate for diagnosing infections due to other serogroups [125]. These tests use monoclonal antibodies that specifically recognize most lipopolysaccharide antigens. According to CDC, approximately 8% of patients with Legionnaires’ disease do not excrete the antigen in their urine [126]. The sensitivity and specificity range differs depending on the study, from 69 to 100% and from 99 to 100%, respectively. Another positive point of UAT is that the results can be available within minutes, following the processing without difficult and expensive instruments. Recently, Ito et al. evaluated a novel test kit that can detect all serogroups of *L. pneumophila* [127].

Moreover, Nakamura et al. reported that new reagents can detect all serogroups by using antibodies that recognize the *L. pneumophila* ribosomal protein L7/L12, in addition to the conventional *L. pneumophila* serogroup 1 lipopolysaccharide [128].

In daily practice, clinical laboratories use the buffered charcoal yeast extract (BCYE) agar, which consists of a CYE agar base supplemented with cysteine, iron salts, and α-ketoglutarate for the growth of *Legionella* spp. in an optimal pH (pH 6.9). In order to isolate *Legionella* spp., there is also another selective media, namely the Glycine-Vancomycin-Polymyxin-Cycloheximide (GVPC) medium. This medium contains antimicrobial agents that inhibit the growth of both Gram-negative and Gram-positive bacteria. In addition, glycine influences the bacterial wall’s permeability, facilitating the action of antibiotics. This agar is usually used for the detection of bacterial strains in environmental samples. The *Legionella* spp. colony grows on culture media, under microaerophilic conditions (2.5% CO_2_). Initially, colonies are small; however, after 4 days of incubation, they increase in diameter (about 3–4 mm). In regard to the colony’s color, it forms white rugose and brown translucent colonies [129].

Recently, matrix-assisted laser desorption/ionization time-of-flight mass spectrometry (MALDI-TOF-MS) has been developed as a rapid and sensitive method for the identification of bacterial species. Given the fact that traditional culture methods lack the sensitivity and specificity for the identification of *Legionella* spp., scientists try to use new more sensitive methods. In this context, Hurst et al., at the end of the 20th century, reported for the first time that MALDI-TOF mass spectrometry is shown to be useful for the detection of *Legionella* spp. [130]. Moliner et al. ran their study to assess the usefulness of MALDI-TOF-MS for rapid species and serogroup identification of *Legionella* spp. The result of this study showed that 94.1% of strains were correctly identified at the species level, while 5.9% were misidentified [131]. Furthermore, He et al. reported that MALDI-TOF mass spectrometry is more sensitive for the identification of *L. pneumophila* than non-*L. pneumophila* strains [132]. The authors suggested that more non-*L. pneumophila* strains need to be included in the MALDI-TOF database in order to increase identification accuracy.

Legionella’s antibodies testing is not helpful for diagnosis. Cross-reactivity between *Legionella* spp. and other microorganisms such as *C. burnetii* has been reported [133]. In addition, Pancer reported that IgM antibodies in sera, collected from children that were hospitalized due to suspected legionellosis, were positive for *L. pneumophila* sgs1–7 and *B. pertussis*. The authors posed a hypothesis of a potential impact of an anti-pertussis vaccination on the results obtained in anti-*L. pneumophila* ELISA IgM tests among children [134]. Sun et al. reported five *L. pneumophila* proteins: FLA, MOMP, Mip, IP, and PILE, which were purified and applied in the serological diagnosis of *L. pneumophila* infections in comparison to a commercial ELISA kit [135]. The five recombinant plasmids pET-fla, pET-momp, pET-mip, pET-ip, and pET-pile were transformed into E. coli BL21. In addition, the proteins were purified by affinity chromatography, tested for the presence of IgG, IgM, and IgA antibodies. For IgG, the sensitivity was 90.4% and the specificity was 97.4%. For IgM, the sensitivity was 91.8% and the specificity was 95.1%. For IgA, the sensitivity was 93.6% and the specificity was 95.3% [135]. It was postulated that the presence of legionella antibodies may be associated with occupational exposure, but there was no evidence of any association with the disease [136].

Recently, Yue et al. used metagenomic, next-generation sequencing (mNGS) technology for the early diagnosis of legionellosis in a patient who rapidly progressed to severe ARDS during the early stages of infection. The specimen from bronchoalveolar lavage fluid (BALF), blood, and urine was negative for infection, as well as for specific anti-Legionella antibodies. Nevertheless, the L. mNGS identification of BALF and blood proved to be the only method that allowed for the detection of *L. pneumophila* [137]. Although the radiological diagnosis of pneumonia includes a chest X-ray and computed tomography [138], pneumonia due to *Legionella* spp. is diagnosed by using the urine antigen test (UAT). However, there were reports in the literature of LP diagnosis with negative UAT results (LPNUAT) based on radiological diagnosis [139]. In patients with suspected legionella pneumonia, a plain chest X-ray in the frontal and lateral projections is recommended. If the localization of the process is unknown, then it is advisable to take an X-ray in the right lateral projection [103]. The revealed changes on the chest X-ray and on the computed tomography in legionella lesions are nonspecific and characteristic of the most atypical pneumonias. The main radiological sign of pneumonia is a decrease in the airiness of the lung tissue due to the accumulation of inflammatory exudate in the respiratory regions. Changes in the lungs along with legionella lesions are more often unilateral and predominantly localized in the lower lobe, but can also appear on both sides, both in the upper and lower parts of the lungs. There may be a small amount of fluid in the pleural cavity. Inflammatory changes in the lungs can be manifested by the following types of infiltration (Figure 3): alveolar type—consolidation of lung tissue with air bronchogram sign (Figure 3A,B), interstitial type—weakly intense compaction of the lung tissue according to the ground-glass opacity (Figure 3C,D), and focal type—multiple polymorphic nodules that are located peribronchially with fuzzy contours (Figure 3E,F). The most characteristic on the chest X-ray and CT in legionella pneumonia are the extensive areas of consolidation of lung tissue of irregular shape, adjacent to the costal and/or horizontal pleura with air bronchogram signs (Figure 4) [140]. (Figure 3 and Figure 4 from Russian Research Center for Radiology and Surgical Technologies of the Russian Federation).

## 7. Conclusions

In recent decades, studies on the surveillance and replication of *L. pneumophila* within human macrophages, ameba, and on the environment have brought insight into the virulence factors and mechanisms employed by the disease. Reports from epidemiological studies have shown remarkable variables in the overall prevalence of LD worldwide [32,141]. One possible explanation may be the use of different diagnostic tests such as PCR, culture, urine antigen test, or MALDI-TOF mass spectrometry techniques. Moreover, the efficiency of the national program for the surveillance of Legionnaires’ disease, the climate, as well as the health system and economic situation of each country are significant factors that affect LD’s prevalence. The environmental surveillance of *Legionella* spp. in the water distribution and cooling systems of healthcare facilities, hotels, and other human-made aquatic systems must be treated as part of a strategy for the prevention of hospital-acquired Legionnaires’ disease.

## Figures and Tables

**Figure 1 jcm-11-06126-f001:**
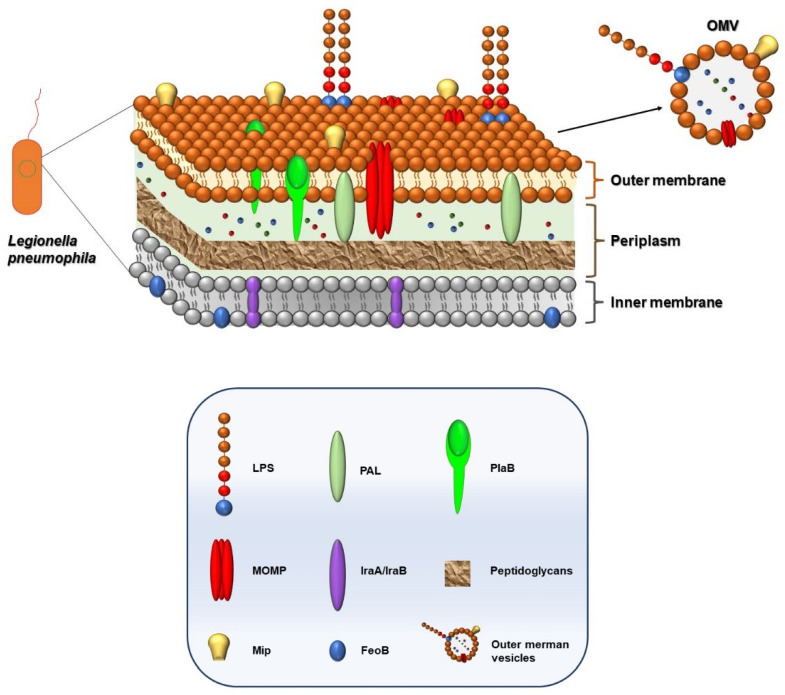
Structure of *Legionella pneumophila*. The outer membrane (OM) is a lipid bilayer composed of phospholipids, lipoproteins, LPS, and proteins, some of which connect the outer membrane to peptidoglycan. In addition, the presence of surface-associated proteins on OM, macrophage infectivity potentiator (Mip) protein, and the major outer membrane protein (MOMP) play a critical role in LD outcome.

**Figure 2 jcm-11-06126-f002:**
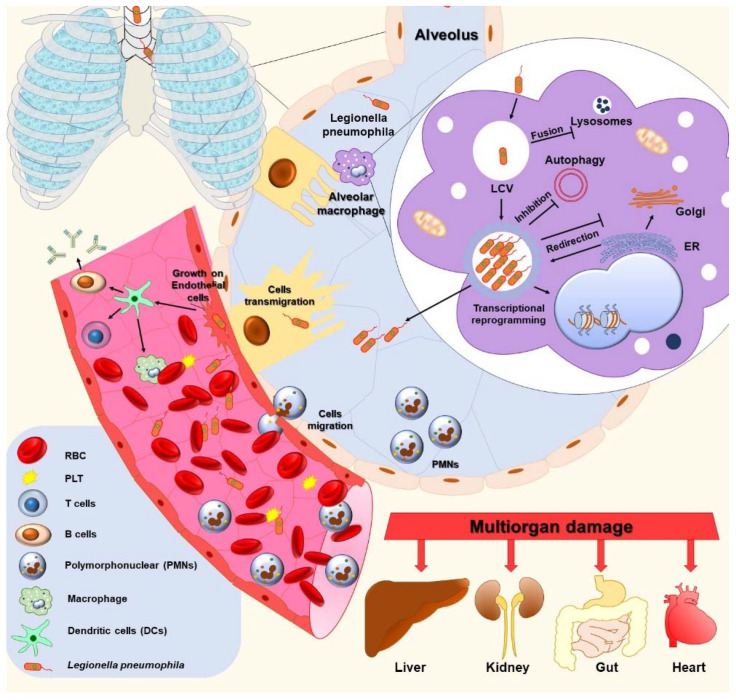
The host cell–*Legionella* spp. interaction and the pathogenesis of LD. Initially, the bacteria adhere to the host cell and are consequently up-taken via phagocytosis, ending within a vacuole (Legionella-containing vacuole (LCV)). Unlike phagosomes, the LCV recruits and redirects different proteins and energy in this specialized vacuole from its own endoplasmic reticulum (ER) and mitochondria. Subsequently, the intracellular replication of *Legionella* spp. is advanced. Second, small GTPase RAB1 secreted into the LCV membrane through the T4SS system results in the redirection of vesicular transport between the ER and the Golgi system. In addition, *L. pneumophila* disposes autophagy inhibitory mechanisms via effector proteins. Finally, bacteria are released into the cytosol and out of the macrophages. Furthermore, *L. pneumophila* binds via Mip protein to the collagen, which allows bacterial transmigration to the blood. *L. pneumophila* induces DCs maturation, which increases secretion of proinflammatory cytokines (IFN γ, TNF-α, IL-6, and IL-1β), which restricts the growth of intracellular Legionella and activates B cells.

**Figure 3 jcm-11-06126-f003:**
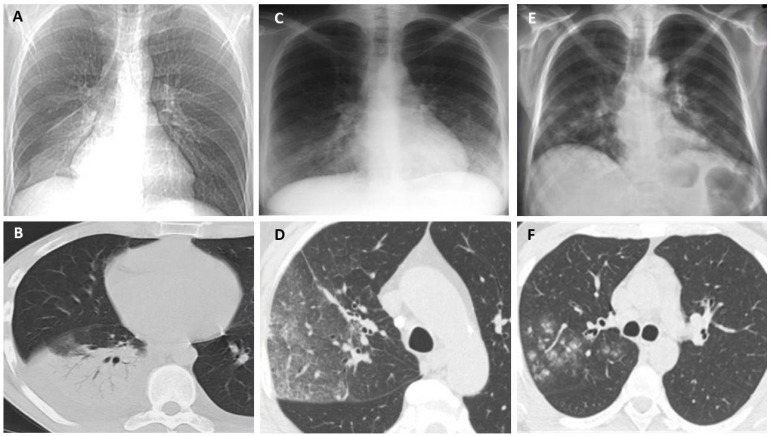
The radiological appearance of bronchopneumonia due to *Legionella* spp. Types of infiltration: Alveolar type—(**A**,**B**); Interstitial type—(**C**,**D**); Focal type—(**E**,**F**).

**Figure 4 jcm-11-06126-f004:**
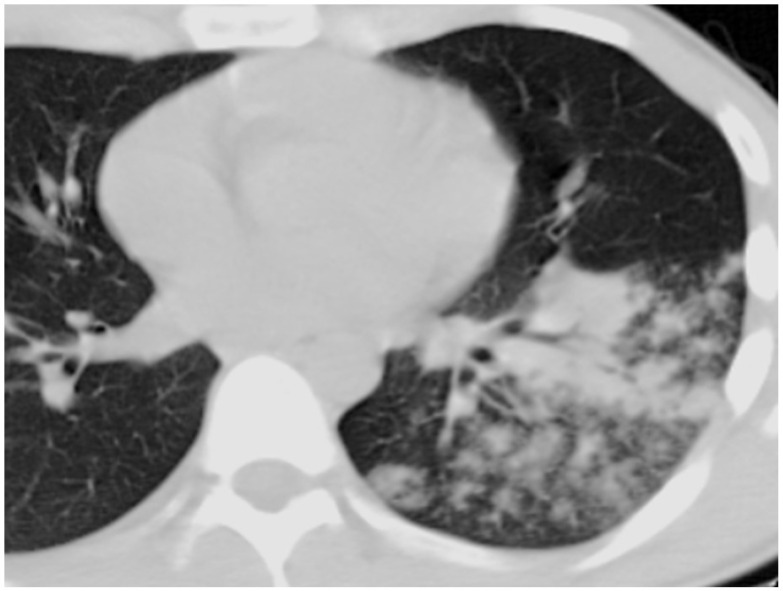
Axial computed tomogram. An extensive area of lung tissue consolidation in the lower lobe of the left lung with air bronchogram signs and with the presence of perifocal alveolar–interstitial ground-glass opacity.

**Table 1 jcm-11-06126-t001:** Virulence factors of *Legionella* spp.

	Virulence Factors	Role	References
Cells’ envelope proteins	FeoB	Attachment to host cells. Iron metabolism	Petermann et al. [55]
	PAL	Activates macrophages Induces cytokine production	Gholipour et al. [57] Shevchuk et al. [53]
	MOMP	Attachment to host cells Inhibits chemotactic activity Modulation of cytokines production	Yang et al. [59]
	LPS	Adhesion to the host cell	Palusinska-Szysz et al. [60]
	Mip	Necessary for intracellular survival	Shen et al. [61]
Other proteins	PilY1	Invasion into non-phagocytic cells Promotes cells’ motility	Hope et al. [62]
	HSP 60	Attachment to host cells Modulation of cytokine expression	Garduño et al. [63]
Secretion systems	T1SS	Secret adhesins, proteins, and enzymes	Qin et al. [64]
	T2SS	Surveillance in lung epithelium cells and macrophage	Tyson et al. [65]
	T3SS	Protein transport mechanism from flagella	Nakono et al. [66]
	T4SS	Organelle trafficking/intracellular growth	Lockwood et al. [67]

## Data Availability

The data in this study are available upon request from the corresponding author.

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
