# Peer review of "Legionella pneumophila: The Journey from the Environment to the Blood"

_jcm, 2022, doi:10.3390/jcm11206126_

Round 1
Reviewer 1 Report
This review by Iliadi et al. represents a collaboration between Russian, Bulgarian and Greek authors. The review has a great title that immediately catches the attention of the reader. Unfortunately, upon reading the paper it is soon realized that its structure, depth and grammatical soundness could be improved. In addition, there seem to be inaccuracies in several of the authors’ scientific statements.
As a reviewer, it has been difficult to mark all the problematic parts of the paper, or make specific suggestions as to how to rewrite/fix those problematic sections. I believe it would not be appropriate to mark all those specific comments in this review as they, simply, are too many. Instead, I have focused my comments on general suggestions to the authors, which I believe could improve their review. The only section of the paper for which I have made specific comments is the Abstract, and that with the sole objective of using it as an example of the problems that I have detected throughout the manuscript.
SPECIFIC COMMENTS ON THE ABSTRACT
Line 22 = Remove “Equally contribution” and use either ‘Equal contribution’ or ‘These authors contributed equally to the paper’.
Line 24 = Since the date of the Legionnaires convention is known (1976) please give the number of years (46) or change to ‘…during the 1976 annual convention of the American Legion…’
Lines 26-27 = Change “Infections” (capitalized) to ‘infections’ (lower case). Also, the sentence contained in these lines needs to be reworded to make the connection between Legionnaires’ disease and Legionella pneumophila. That is, authors need to indicate here that L. pneumophila is the causal agent of Legionnaires’ disease. Finally, infections of humans with L. pneumophila can also occur by aspiration of contaminated water, meaning that the authors’ statement that infections “arise exclusively by inhalation of aerosols” is not clinically accurate.
Lines 27-29 = The sentences contained in these lines also have several problems. When authors refer to Legionella pneumophila (which is one species within the genus) they cannot say that it contains 32 Legionella species! Perhaps the authors are referring to the genus Legionella, but that does not seem to be the case either, because there are more than 35 species in the genus. So, it is unclear what the authors are talking about. Consequently, when the authors refer to the 51 serogroups, it is not clear whether they mean Legionella serogroups, or L. pneumophila serogroups. This needs clarification. In addition, it should be indicated that L. pneumophila strains belonging to serogroup 1 (Lp1) are the most frequently detected in the clinic. This is important because in the freshwater environment (natural or man-made) Lp1 might not be the most frequently detected serogroup, depending on geographical location.
Line 32 = “Cell-to-cell spreading”. This term is conventionally used in relation to intracellular pathogens that pass from one cell to another without leaving the intracellular environment (like Listeria monocytogenes or Shigella spp.). Please use a different term, or simply say ‘spreading’. Also, please add ‘invasion of host cells,’ after “…attachment to host cells,” and remove the word “and” before “intracellular replication”.
Lines 37-38 = The sentence contained in these lines needs to be moved up, immediately after the sentence that ends “…personal immune status of the patient”.
GENERAL COMMENTS ON THE MANUSCRIPT
By publishing in the Journal of Clinical Medicine, the general focus of the review should be on clinical aspects of legionellosis, and in particular on ‘the journey of Legionella pneumophila from the environment to the blood’. As mentioned above, the title of the review is very good and enticing, so the authors should use it to their advantage, and focus on discussing in depth the clinical aspects of legionellosis, and how L. pneumophila manages to enter the human body through the respiratory tract, enter then the bloodstream, and spread to other organs and systems.
The authors should drop the microbiological/ecological, as well as the molecular pathogenesis aspects of legionellosis, and just provide (for the reader’s sake) references to recent comprehensive reviews on those subjects. Currently, the sections on ‘Monitoring of L. pneumophila in the environment’, ‘Molecular mechanisms of LD’ and ‘Virulence Factors of L. pneumophila’ add little to the main topic of the review, so they should be deleted. The remaining topics of the review (Epidemiology of LD, Cellular aspects of LD, Clinical manifestations of legionellosis, and Clinical diagnosis) should be reviewed more in depth, mainly because these subjects are currently treated very superficially. For instance, in relation to the Epidemiology of LD, the authors (who are all European) should either focus on the European epidemiology of LD and cover this topic with more detail, or alternatively focus on the global epidemiology of LD, including then more comprehensive up to date data from Europe, America, Australia, and Asian and African countries. In particular, it would be very important for the authors to provide a comprehensive and up to date view of the transmigration of L. pneumophila across the lung epithelium to reach the internal connective tissue, enter the bloodstream, and reach several internal organs, which is a topic not commonly covered in reviews concerning Legionnaires’ disease.
The references (citations) should be thoroughly revised, because there are some which do not seem to correspond to the statements where they are used. For instance (just as an example) the reference to Hsp60 in Table 1 (Garduno et al. [72]) seems adequate to support the role of Hsp60 in attachment to host cells and invasion, but it is not the right reference to support the role of Hsp60 in the modulation of cytokine expression. Some other citations have typographical errors like that in line 309 (also mentioned just as an example), where it should be given as Wagner (instead of Wanger).
Finally, it would be necessary for the authors to thoroughly revise the manuscript to correct its grammar. Contacting an individual proficient in the English language, who is also knowledgeable in Clinical Microbiology/Infectious Diseases, would be highly desirable for the proper revision of the manuscript.
Author Response
We would like to thank you deeply for all your comments and for the time you spent in our article. All your corrections and comments considered valuable and empowered us to improve the quality of our manuscript.
All corrections in the manuscript were performed by track changes of words.
Comments and Suggestions for Authors
This review by Iliadi et al. represents a collaboration between Russian, Bulgarian and Greek authors. The review has a great title that immediately catches the attention of the reader. Unfortunately, upon reading the paper it is soon realized that its structure, depth and grammatical soundness could be improved. In addition, there seem to be inaccuracies in several of the authors’ scientific statements.
As a reviewer, it has been difficult to mark all the problematic parts of the paper, or make specific suggestions as to how to rewrite/fix those problematic sections. I believe it would not be appropriate to mark all those specific comments in this review as they, simply, are too many. Instead, I have focused my comments on general suggestions to the authors, which I believe could improve their review. The only section of the paper for which I have made specific comments is the Abstract, and that with the sole objective of using it as an example of the problems that I have detected throughout the manuscript.
SPECIFIC COMMENTS ON THE ABSTRACT
Line 22 = Remove “Equally contribution” and use either ‘Equal contribution’ or ‘These authors contributed equally to the paper’.
Response
It was corrected as Equal contribution
Line 24 = Since the date of the Legionnaires convention is known (1976) please give the number of years (46) or change to ‘…during the 1976 annual convention of the American Legion…’
Response
Done
Lines 26-27 = Change “Infections” (capitalized) to ‘infections’ (lower case).
Response
Done
Also, the sentence contained in these lines needs to be reworded to make the connection between Legionnaires’ disease and Legionella pneumophila. That is, authors need to indicate here that L. pneumophila is the causal agent of Legionnaires’ disease. Finally, infections of humans with L. pneumophila can also occur by aspiration of contaminated water, meaning that the authors’ statement that infections “arise exclusively by inhalation of aerosols” is not clinically accurate.
Response
Done
Lines 27-29 = The sentences contained in these lines also have several problems. When authors refer to Legionella pneumophila (which is of the text one species within the genus) they cannot say that it contains 32 Legionella species! Perhaps the authors are referring to the genus Legionella, but that does not seem to be the case either, because there are more than 35 species in the genus. So, it is unclear what the authors are talking about. Consequently, when the authors refer to the 51 serogroups, it is not clear whether they mean Legionella serogroups, or L. pneumophila serogroups. This needs clarification.
Response
This paragraph was modified: “The family Legionellaceae and the genus Legionella is the causal agent of LD. The genus Legionella is a genetically heterogeneous species, composed of a numerously species and serogroups”
In addition, it should be indicated that L. pneumophila strains belonging to serogroup 1 (Lp1) are the most frequently detected in the clinic. This is important because in the freshwater environment (natural or man-made) Lp1 might not be the most frequently detected serogroup, depending on geographical location.
Response
We agree with the reviewer and write that this strain is one of the most detected strains that causes outbreaks worldwide. Authors are not tried to support that Lp1 is the most frequently detected serogroup of the environment (natural or man-made).
To avoid this, we specified that Lp1is the most detected clinical strains.
Line 32 = “Cell-to-cell spreading”. This term is conventionally used in relation to intracellular pathogens that pass from one cell to another without leaving the intracellular environment (like Listeria monocytogenes or Shigella spp.). Please use a different term, or simply say ‘spreading’. Also, please add ‘invasion of host cells,’ after “…attachment to host cells,” and remove the word “and” before “intracellular replication”.
Response
Moreover, some corrections were done”
Line 36 “activates the” was changed to “and”, Line 38 L. pneumophila corrected to italics and “a” changed to “the”.
Lines 37-38 = The sentence contained in these lines needs to be moved up, immediately after the sentence that ends “…personal immune status of the patient”.
Response
The sentence “The infection has two distinct forms: non- pneumatic form or Pontiac Fever, which is a milder febrile flu-like illness and LD, a more severe form of infection which includes pneumonia. Moreover, there were reports of extrapulmonary manifestations of LD such as cardiac, brain, abdominal, and joints involvement” was moved up.
GENERAL COMMENTS ON THE MANUSCRIPT
By publishing in the Journal of Clinical Medicine, the general focus of the review should be on clinical aspects of legionellosis, and in particular on ‘the journey of Legionella pneumophila from the environment to the blood’. As mentioned above, the title of the review is very good and enticing, so the authors should use it to their advantage, and focus on discussing in depth the clinical aspects of legionellosis, and how L. pneumophila manages to enter the human body through the respiratory tract, enter then the bloodstream, and spread to other organs and systems.
The authors should drop the microbiological/ecological, as well as the molecular pathogenesis aspects of legionellosis, and just provide (for the reader’s sake) references to recent comprehensive reviews on those subjects. Currently, the sections on ‘Monitoring of L. pneumophila in the environment’, ‘Molecular mechanisms of LD’ and ‘Virulence Factors of L. pneumophila’ add little to the main topic of the review, so they should be deleted. The remaining topics of the review (Epidemiology of LD, Cellular aspects of LD, Clinical manifestations of legionellosis, and Clinical diagnosis) should be reviewed more in depth, mainly because these subjects are currently treated very superficially. For instance, in relation to the Epidemiology of LD, the authors (who are all European) should either focus on the European epidemiology of LD and cover this topic with more detail, or alternatively focus on the global epidemiology of LD, including then more comprehensive up to date data from Europe, America, Australia, and Asian and African countries. In particular, it would be very important for the authors to provide a comprehensive and up to date view of the transmigration of L. pneumophila across the lung epithelium to reach the internal connective tissue, enter the bloodstream, and reach several internal organs, which is a topic not commonly covered in reviews concerning Legionnaires’ disease.
The references (citations) should be thoroughly revised, because there are some which do not seem to correspond to the statements where they are used. For instance (just as an example) the reference to Hsp60 in Table 1 (Garduno et al. [72]) seems adequate to support the role of Hsp60 in attachment to host cells and invasion, but it is not the right reference to support the role of Hsp60 in the modulation of cytokine expression. Some other citations have typographical errors like that in line 309 (also mentioned just as an example), where it should be given as Wagner (instead of Wanger).
Finally, it would be necessary for the authors to thoroughly revise the manuscript to correct its grammar. Contacting an individual proficient in the English language, who is also knowledgeable in Clinical Microbiology/Infectious Diseases, would be highly desirable for the proper revision of the manuscript.
Response
You can follow all the corrections made, in the text, with word trach changes.
Reviewer 2 Report
This paper is an interesting synthesis about the capacity of L. pneumophila to contaminate people from the environment, the various clinical forms, and the methods of diagnosis. The major modifications needed to improve the review concern the citations. Some references were not properly cited and need to be corrected (see detailed comments below). Throughout the document the bacterial names are sometimes not in italic (e.g. L. 35), so please rectify.
L. 24 & 44: place “USA” into brackets.
L. 25: replace the point by a comma between “appeared” and “and”.
L.26: remove the uppercase at “Infection”
L. 36 and 152: replace “bacteriemia” replaced by “bacteremia”
L. 57: I suggest to replace “L. pneumophila have mechanisms of the mobility” by “L. pneumophila is a motile bacterium.”
L. 57: replace “showed” by “shown”
L. 60: Reference 8 does not justify the loss of multiplication at 46°C. Mauchline et al. described the loss of flagella. You could cite for example: Konishi, T., Yamashiro, T., Koide, M., Nishizono, A., 2006. Influence of temperature on growth of Legionella pneumophila biofilm determined by precise temperature gradient incubator. Journal of Bioscience and Bioengineering 101, 478–484. https://doi.org/10.1263/jbb.101.478. The authors have demonstrated the absence of multiplication above 44°C.
L. 73: add “and to” between spp. and relate
L. 80-82: “Potential sources of Legionella spp transmission are potable water sources: showers and taps, and non-potable sources: fountains, spas, cooling towers and evaporative condensers [14].” This sentence comes from reference 14, but it is not the source of the information: “Potential sources of Legionella transmission include potable water sources, such as fountains, showers and taps, and non-potable sources such as spas, cooling towers and evaporative condensers (Steinert et al., 2007; Newton et al., 2010; Cunha et al., 2016).”
L. 85: I suggest to replace “may comprise reservoir” by “may constitute a reservoir”
L. 86: the sentence is not clear, I suggest: “Oda et al. reported the detection of L. pneumophila…”
L. 87: in a 62-year-old man who
L. 155: remove the point between “thesis” and “[49, 50]”
L. 160-161: “L. pneumophila encodes a variety of proteins similar in functionality to eukaryotic proteins”: this sentence is not supported by reference [52] but rather by Gomez-Valero, L., Rusniok, C., Jarraud, S., Vacherie, B., Rouy, Z., Barbe, V., Medigue, C., Etienne, J., Buchrieser, C., 2011. Extensive recombination events and horizontal gene transfer shaped the Legionella pneumophila genomes. BMC Genomics 12, 536. https://doi.org/10.1186/1471-2164-12-536
L. 183: replace “inactivates” by “inactivation”
L. 188-190: “The periplasm, based on its property, contains soluble enzymes, such as proteases and nucleases has been considered an evolutionary precursor of the classical lysosomes of the eukaryotic cells [60].” The sentence should be rewritten and the original author cited: “the periplasm has been called an evolutionary precursor of the lysosomes of eukaryotic cells (De Duve and Wattiaux, 1966).”
L. 190-192: there is no reference for this sentence “Moreover, the periplasm contains one of the most important substances, a peptidoglycan, that plays a consequential role as a microbial virulence factor and contributes to the survival of macrophages and amoeba.” And I guess that the idea is that this peptidoglycan contributes to the survival of L. pneumophila inside macrophages and amoeba. Is it correct?
L. 192-194: “Furthermore, in this space, there are several co-localized antigens, including peptidoglycan associated lipoprotein (PAL) protein [61].” The reference [61] described the cloning of pal gene in E. coli and does not attest of the presence of this molecule in the periplasm, so this reference is not appropriated. This remark also applies to reference for PAL role in Table 1.
L. 210: “Yang at el » => Yang et al.
L. 220-221 “Mip could influence macrophages on phagocytic and chemotactic activities through different arises such as miR-21 [65].” Do the authors mean axis as in reference [65]: “The present results revealed that MIP could influence RAW264.7 macrophages on phagocytic and chemotactic activities through the axis of lncRNA GAS5/miR-21/SOCS6.”?
L. 224-225: “Interestingly, the targeting on the Mip gene has been recognized as the only useful tool for Legionella non-pneumophila species identification in samples [67].” In the reference [67], the authors explained that mip gene alone is not sufficient and that both the mip and rpoB genes makes it possible to correctly discriminate between several species. Please rectify.
Table 1: why is it written below the title “The type I secretion system (T1SS)”? I suggest to remove it.
L. 279: Dot/Icm encoding genes or dot/icm genes
L. 323: Please write dendritic cells in full before using DCs
L. 361-362: reference [107] is not properly cited. It demonstrates the efficacy of PCR compared to culture to identify L. pneumophila. The symptoms of the patient are described but there is no mention of myalgia or confusion in that case. Please cite the proper reference.
L. 368: “(presenting)” should be suppressed.
L.376-382: What is the origin (publication? Website? Report?) of the death percentages cited? It should be reported. Reference [112] L. 383 corresponds only to the last sentence.
L. 388-390: The references corresponding to “follow-up clinical studies” should be cited.
L.394: As there is no 4.2., I propose to remove the 4.1. title or to add titles to previous paragraphs in part 4.
L.410-411: “legionellosis can often be associated with neurological symptoms, which is an indication of hematogenic complication”. The cited reference justifies that legionellosis can be associated with neurological symptoms, but I am not sure that it is an indication of hematogenic complication. Could you please comment and/or justify this assertion?
L.440-448: The paragraph should be grouped with previous sentence L. 423-424 as it is the same assertion. In the same way L. 448-450 correspond to L. 424-427.
L. 460-463: Please rectify the percentages according to reference [131]: “for IgG the sensitivity was 90.4%, the specificity was 97.4%. For IgM the sensitivity was 91.8%, the specificity was 95.1%. For IgA the sensitivity was 93.6%, the specificity was 95.3%.”
L. 463-465: The authors of reference [132] did not say that legionella antibody presence “is usually associated with occupational exposure”, but rather that it “may be associated”.
L. 470-471: I guess that the idea was to say that mNGS was the only method that allowed the detection of L. pneumophila. As written, the reader will understand that L. pneumophila was the only microorganism detected by mNGS. Which one is the intended idea? Please modify the sentence if the conclusion is the first one.
L. 472: As previously, there is no 5.2. in this part, so I recommend to remove this subtitle or to add a subtitle at the beginning of part 5.
L. 481-493: there is no reference associated with these sentences. Please justify.
L.496-498: I don’t understand the sentence. Is the verb “assisted” properly used? The beginning of the sentence L. 498 lacks or is it the end of previous sentence?
L. 501-502: the detection of the disease by MALDI-TOF is not reported in paragraph 5. It should be added.
Reference [45] is not properly written. The website to access the pdf should be cited.
Figure 2 is not cited in the document. Several sentences of the legend should be corrected (e.g. to allowing)
Figure 3 and 4: What is the origin of the radiographies?
Author Response
Dear reviewers,
We would like to thank you deeply for all your comments and for the time you spent in our article. All your corrections and comments considered valuable and empowered us to improve the quality of our manuscript.
All corrections in the manuscript were performed by track changes of words.
This paper is an interesting synthesis about the capacity of L. pneumophila to contaminate people from the environment, the various clinical forms, and the methods of diagnosis. The major modifications needed to improve the review concern the citations. Some references were not properly cited and need to be corrected (see detailed comments below). Throughout the document the bacterial names are sometimes not in italic (e.g. L. 35), so please rectify.
Response
Done
- 24 & 44: place “USA” into brackets.
Response
Done
- 25: replace the point by a comma between “appeared” and “and”.
Response
Done
L.26: remove the uppercase at “Infection”
Response
Done
- 36 and 152: replace “bacteriemia” replaced by “bacteremia”
Response
Done
- 57: I suggest to replace “L. pneumophilahave mechanisms of the mobility” by “L. pneumophilais a motile bacterium.”
Response
Done
- 57: replace “showed” by “shown”
Response
Done
- 60: Reference 8 does not justify the loss of multiplication at 46°C. Mauchline et al. described the loss of flagella. You could cite for example: Konishi, T., Yamashiro, T., Koide, M., Nishizono, A., 2006. Influence of temperature on growth of Legionella pneumophila biofilm determined by precise temperature gradient incubator. Journal of Bioscience and Bioengineering 101, 478–484. https://doi.org/10.1263/jbb.101.478. The authors have demonstrated the absence of multiplication above 44°C.
Response
Reference 8 "Physiology and morphology of Legionella pneumophila in continuous culture at low oxygen concentration" by Maunchline et al was replaced by article of Konishi T. et al "Influence of Temperature on Growth of Legionella Pneumophi-la Biofilm Determined by Precise Temperature Gradient Incubator." as reviewers’ recommended.
- 73: add “and to” between spp. and relate
Response
“and” was added
- 80-82: “Potential sources of Legionella spp transmission are potable water sources: showers and taps, and non-potable sources: fountains, spas, cooling towers and evaporative condensers [14].” This sentence comes from reference 14, but it is not the source of the information: “Potential sources of Legionellatransmission include potable water sources, such as fountains, showers and taps, and non-potable sources such as spas, cooling towers and evaporative condensers (Steinert et al., 2007; Newton et al., 2010; Cunha et al., 2016).”
Response
This sentence was modified as “Potential sources of Legionella transmission include potable water sources, such as fountains, showers and taps, and non-potable sources such as spas, cooling towers and evaporative condensers. Moreover reference 14 has changed to “Cunha BA, Burillo A, Bouza E. Legionnaires' disease. 2016”
- 85: I suggest to replace “may comprise reservoir” by “may constitute a reservoir”
Response
Done
- 86: the sentence is not clear, I suggest: “Oda et al. reported the detection of L. pneumophila…”
Response
Done
- 87: in a 62-year-old man who
Response
Done
- 155: remove the point between “thesis” and “[49, 50]”
Response
Done
- 160-161: “L. pneumophila encodes a variety of proteins similar in functionality to eukaryotic proteins”: this sentence is not supported by reference [52] but rather by Gomez-Valero, L., Rusniok, C., Jarraud, S., Vacherie, B., Rouy, Z., Barbe, V., Medigue, C., Etienne, J., Buchrieser, C., 2011. Extensive recombination events and horizontal gene transfer shaped the Legionella pneumophila genomes. BMC Genomics 12, 536. https://doi.org/10.1186/1471-2164-12-536
Response
Reference 52 has changed to Gomez-Valero et al.
- 183: replace “inactivates” by “inactivation”
Response
Done (line 192in new version of article)
- 188-190: “The periplasm, based on its property, contains soluble enzymes, such as proteases and nucleases has been considered an evolutionary precursor of the classical lysosomes of the eukaryotic cells [60].” The sentence should be rewritten and the original author cited: “the periplasm has been called an evolutionary precursor of the lysosomes of eukaryotic cells (De Duve and Wattiaux, 1966).”
Response
This sentence was rephrased. Moreover, the original article (de Duve, C.; Wattiaux, R. Functions of Lysosomes. Annual Review of Physiology 1966) was cited.
- 190-192: there is no reference for this sentence “Moreover, the periplasm contains one of the most important substances, a peptidoglycan, that plays a consequential role as a microbial virulence factor and contributes to the survival of macrophages and amoeba.” And I guess that the idea is that this peptidoglycan contributes to the survival of L. pneumophilainside macrophages and amoeba. Is it correct?
Response
Done
- 192-194: “Furthermore, in this space, there are several co-localized antigens, including peptidoglycan associated lipoprotein (PAL) protein [61].” The reference [61] described the cloning of palgene in E. coliand does not attest of the presence of this molecule in the periplasm, so this reference is not appropriated. This remark also applies to reference for PAL role in Table 1.
Response
One other article (Shevchuk et al) that describe several co-localized proteins was added.
- 210: “Yang at el » => Yang et al.
Response
Done
- 220-221 “Mip could influence macrophages on phagocytic and chemotactic activities through different arises such as miR-21 [65].” Do the authors mean axis as in reference [65]: “The present results revealed that MIP could influence RAW264.7 macrophages on phagocytic and chemotactic activities through the axis of lncRNA GAS5/miR-21/SOCS6.”?
Response
This sentence was rephrased: “Shen et al, reports that Mip could influence macrophages on phagocytic and chemotactic activities. It was shown, on murine macro-phage cells line (RAW264.7), that MIP increases the cells' chemotaxis and inhibits the phagocytosis”
- 224-225: “Interestingly, the targeting on the Mip gene has been recognized as the only useful tool for Legionella non-pneumophila species identification in samples [67].” In the reference [67], the authors explained that mipgene alone is not sufficient and that both the mipand rpoB genes makes it possible to correctly discriminate between several species. Please rectify.
Response
It is widely recognized that non-Lp species can be identified by only the mip gene. In contrary, several studies have shown that to identify L. pneumophila strains needs to target more than one gene. No single gen target system is perfect to identify L. pneumophila.
Table 1: why is it written below the title “The type I secretion system (T1SS)”? I suggest to remove it.
Response
Done
- 279: Dot/Icm encoding genes or dot/icm genes
Response
Done
- 323: Please write dendritic cells in full before using DCs
Response
Done
- 361-362: reference [107] is not properly cited. It demonstrates the efficacy of PCR compared to culture to identify L. pneumophila. The symptoms of the patient are described but there is no mention of myalgia or confusion in that case. Please cite the proper reference.
Response
Done
- 368: “(presenting)” should be suppressed.
Response
Done
L.376-382: What is the origin (publication? Website? Report?) of the death percentages cited? It should be reported. Reference [112] L. 383 corresponds only to the last sentence.
Response
New citations were added
- 388-390: The references corresponding to “follow-up clinical studies” should be cited.
Response
This sentence was rephrased. New references were added.
L.394: As there is no 4.2., I propose to remove the 4.1. title or to add titles to previous paragraphs in part 4.
Response
We re-organized the paragraphs.
L.410-411: “legionellosis can often be associated with neurological symptoms, which is an indication of hematogenic complication”. The cited reference justifies that legionellosis can be associated with neurological symptoms, but I am not sure that it is an indication of hematogenic complication. Could you please comment and/or justify this assertion?
Response
Indeed, only few cases of CNS complications due to Lp with cerebellar symptoms have been reported in the literature.
We agree with reviewer that just present of neurological symptoms, ist'nt strong indication of hematogenic complication.
Hibino et al, report a case of Legionnaires' pneumonia accompanied by encephalitis. Unfortunately, this case report has written in Japanese. For this reason, this article was not included in our review.
L.440-448: The paragraph should be grouped with previous sentence L. 423-424 as it is the same assertion. In the same way L. 448-450 correspond to L. 424-427.
Response
All paragraphs were grouped as reviewer proposed.
- 460-463: Please rectify the percentages according to reference [131]: “for IgG the sensitivity was 90.4%, the specificity was 97.4%. For IgM the sensitivity was 91.8%, the specificity was 95.1%. For IgA the sensitivity was 93.6%, the specificity was 95.3%.”
Response
Done
- 463-465: The authors of reference [132] did not say that legionella antibody presence “is usually associated with occupational exposure”, but rather that it “may be associated”.
Response
Done
- 470-471: I guess that the idea was to say that mNGS was the only method that allowed the detection of L. pneumophila. As written, the reader will understand that L. pneumophilawas the only microorganism detected by mNGS. Which one is the intended idea? Please modify the sentence if the conclusion is the first one.
Response
We rephrased this sentence
- 472: As previously, there is no 5.2. in this part, so I recommend to remove this subtitle or to add a subtitle at the beginning of part 5.
Response
Done
- 481-493: there is no reference associated with these sentences. Please justify.
Response
Done
L.496-498: I don’t understand the sentence. Is the verb “assisted” properly used? The beginning of the sentence L. 498 lacks or is it the end of previous sentence?
Response
The sentence was rephrased as “Over the last few decades, studies assisted in the surveillance and replication of L pneumophila within human macrophages, ameba and on the environmental have brought insight into the virulence factors and mechanisms employed by the disease.”
- 501-502: the detection of the disease by MALDI-TOF is not reported in paragraph 5. It should be added.
Response
New paragraph was added: “Recently, matrix-assisted laser desorption/ionization time-of-flight mass spectrometry (MALDI-TOF-MS) has developed as a rapid and sensitive method for identification of bacterial species. Given the fact that traditional culture methods lack the sensitivity and specificity for identification of Legionella spp, scientists try to use new more sensitive meth-ods. In this context, Hurst et al, at the end of 20 century report for the first time that MAL-DI-TOF mass spectrometry is shown to be useful for detection of Legionella spp [133]. Mo-liner et al, run their study to assess the usefulness of MALDI-TOF-MS for rapid species and serogroup identification of Legionella spp. The result of this study shown that 94.1 % strains were correctly identified at the species level, while 5.9 % were misidentified [134]. Furthermore, He et al, report that MALDI-TOF mass spectrometry is more sensitive for identification of L. pneumophila than non-L. pneumophila strains [135]. Authors suggest that more non-L. pneumophila strains, need to be included in the MALDI-TOF database to in-crease identification accuracy.”
Reference [45] is not properly written. The website to access the pdf should be cited.
Response
Reference 45 was deleted.
We try to add the website to access the pdf "https://eody.gov.gr/wp-content/uploads/2019/01/EWGLI-Technical-Guidelines-legeonela.pdf" but unfortunately zotero programm adds only the title.
Figure 2 is not cited in the document. Several sentences of the legend should be corrected (e.g. to allowing)
Response
Figure 2 unintentionally was cited as Figure 1.
We correct this mistake.
Figure 3 and 4: What is the origin of the radiographies?
Response
All radiological figures that presented in this paper are derived from archives of Russian Research Center for Radiology and Surgical Technologies of the Russian Federation. All patients have signed a written consent to give the permission for publication of the radiological figures.
Reviewer 3 Report
In this present review, authors have discussed that Legionella pneumophila is the environmental microorganism. The clinical outbreak of Legionnaires’s disease (LD) depends upon bacterial load in aerosol, the virulence factors and the status of immune system of a patient. The process of pathogenesis of LD starts with bacterial cell attachment to host cells, its intracellular replication and cell to cell spreading through translocation of proteins using type IV secretion system. The transmigration of LD to lung epithelial cells leads to bacteriemia, septic shock and multiorgan complications.
Comments:
1. This review about Legionella pneumophila is well explained and it has discussed the basic information regarding epidemiology; the virulence factors and their mechanism; diagnosis criteria and clinical manifestation of Legionnaires’s disease. However, as the title suggests “environment to blood”; the information regarding the Legionella pneumophila infection in blood (Bacteriemia) is very less. Please discuss this in detail.
2. Emergence of LD due to water stagnation in prolonged building closures is situation based. This can happen in any case and linking this with COVID-19 is not justified.
3. Please change ‘efficient inactivates of macrophages’ to ‘efficient inactivation of macrophages’ (Page 4; Line 183)
4. Please check ‘literature of Legionellosis with associated rash’ that can be written as ‘literature of Legionellosis association with rash’. (Page 8; Line 350)
Author Response
Dear reviewers,
We would like to thank you deeply for all your comments and for the time you spent in our article. All your corrections and comments considered valuable and empowered us to improve the quality of our manuscript.
All corrections in the manuscript were performed by track changes of words.
- This review about Legionella pneumophila is well explained and it has discussed the basic information regarding epidemiology; the virulence factors and their mechanism; diagnosis criteria and clinical manifestation of Legionnaires’s disease. However, as the title suggests “environment to blood”; the information regarding the Legionella pneumophilainfection in blood (Bacteriemia) is very less. Please discuss this in detail.
Response
We tried to include all article about bacteremia and LD in separate section with same name “Bacteremia”
- Emergence of LD due to water stagnation in prolonged building closures is situation based. This can happen in any case and linking this with COVID-19 is not justified.
Response
We agree with the reviewer that prolonged building closures and stagnation in building water systems during the COVID-19 pandemic contribute to appearance COVID-19 and legionella co-infection.
Moreover, cases with Community-acquired pneumonia due to legionella spp and COVID-19 are reported.
In this context, we decide to include the section "The impact of the Coronavirus Disease (COVID-19) pandemic to Legionella spp. epidemic" to highlight the importance of the detection accuracy pathogens, in the time of COVID‐19 pandemic to start antimicrobial therapy without delay.
- Please change ‘efficient inactivates of macrophages’ to ‘efficient inactivation of macrophages’ (Page 4; Line 183)
Response
Done
- Please check ‘literature of Legionellosis with associated rash’ that can be written as ‘literature of Legionellosis association with rash’. (Page 8; Line 350)
Response
Done
Reviewer 4 Report
This review paper, " Legionella pneumophila: the journey from the environment to 2 the blood,” tries to perform a comprehensive literature search on LD and its pathogenesis. However, the writing of the paper is currently incoherent. The flow of the paper can be significantly improved. Some of the major comments I have are listed below.
The abstract of the paper is taken from the introduction and there appears to be a lot of repetition. Please consider rewriting the abstract with the idea that the abstract should be able to summarize the entire paper and what exactly is being reviewed in the paper. This will help draw the attention of readers to the paper.
Please italicize the name of the bacterium throughout the paper. There appear to be inconsistencies.
Spelling mistakes at several places – For example Line 108 titer? Line 177: physiochemical?
Section 2.1 is an attempt to suggest the impact of Covid 19 on LD. However, this section is not clear and doesn’t provide any conclusive evidence of any impact. The writing of the section can be improved or can be completely omitted from the manuscript
Section 2.2 – why is “an” in red?
The headings of these sections can also be changed to clearly reflect what is going on in the paragraph
Lines 153-156: Please rephrase. “Lp handles host cells” is colloquial
Line 182-183: “in vitro 182 studies showed that FeoB is required for efficient inactivates of macrophages” - This is unclear to me
Lines 187-188: Defining periplasm may be unnecessary – “The periplasmic space or Periplasm is a gel-like substance located between the outer 187 and inner membranes.”
Table 1: References needs to be formatted in the correct way
Lines 250-252: Please rephrase – “During the stationary phase 250 of life, after the bacteria has stopped replicating, L. pneumophila secrete a molecule called 251 homogenetisic acid (HGA), which is produced by amino acids’ tyrosine and phenylalanine. “How is it produced?
Figures 1 and 2. It is not clear if the figures are originally generated or taken from a source
Figures 3 and 4: Please indicate the source of these figures
Author Response
Dear reviewers,
We would like to thank you deeply for all your comments and for the time you spent in our article. All your corrections and comments considered valuable and empowered us to improve the quality of our manuscript.
All corrections in the manuscript were performed by track changes of words.
The abstract of the paper is taken from the introduction and there appears to be a lot of repetition. Please consider rewriting the abstract with the idea that the abstract should be able to summarize the entire paper and what exactly is being reviewed in the paper. This will help draw the attention of readers to the paper.
Please italicize the name of the bacterium throughout the paper. There appear to be inconsistencies.
Response
Done
Spelling mistakes at several places – For example Line 108 titer? Line 177: physiochemical?
Response
Done
Section 2.1 is an attempt to suggest the impact of Covid 19 on LD. However, this section is not clear and doesn’t provide any conclusive evidence of any impact. The writing of the section can be improved or can be completely omitted from the manuscript
Response
We decide to include the section "The impact of the Coronavirus Disease (COVID-19) pandemic to Legionella spp. epidemic" to highlight the importance of the detection accuracy pathogens, in the time of COVID‐19 pandemic to start antimicrobial therapy without delay.
Section 2.2 – why is “an” in red?
Response
Red color was deleted.
Lines 153-156: Please rephrase. “Lp handles host cells” is colloquial
Response
This sentence was rephrased
“Following entry, L. pneumophila influences host cells’ function to create an environment that protects them against the host’s immune mechanisms and provides nutrition for growth”
Line 182-183: “in vitro 182 studies showed that FeoB is required for efficient inactivates of macrophages” - This is unclear to me
Response
The uptake of Fe by Legionella is mainly carried out by the FeoB. Therefore, the cellular concentration of Fe is reduced. Nevertheless, iron plays an important role in macrophage’s function, and their differentiation into M1 and M2. Iron deficiency into macrophages can also affect their transcriptional profiles, including a distinct regulation of genes related to their antimicrobial activity.
Lines 187-188: Defining periplasm may be unnecessary – “The periplasmic space or Periplasm is a gel-like substance located between the outer 187 and inner membranes.”
Response
This sentence was deleted.
Table 1: References needs to be formatted in the correct way
Response
In this article we use program zotero, that recommended and trusted by scientist worldwide, to design references.
References on the table are formatted by the same manner that in the text.
Lines 250-252: Please rephrase – “During the stationary phase 250 of life, after the bacteria has stopped replicating, L. pneumophila secrete a molecule called 251 homogenetisic acid (HGA), which is produced by amino acids’ tyrosine and phenylalanine. “How is it produced?
Response
This sentence was rephrased
Figures 1 and 2. It is not clear if the figures are originally generated or taken from a source
Response
Figures 1 and 2 are original and were completely generated by authors (TK- Theocharis Konstantinidis).
Figures 3 and 4: Please indicate the source of these figures
Response
All radiological figures that presented in this paper from archive of Russian Research Center for Radiology and Surgical Technologies of the Russian Federation. All patients have signed a written consent to give the permission for publication of the radiological figures.
Round 2
Reviewer 1 Report
The revised manuscript by Iliadi et al. is basically the same as in the original submission.
The authors ignored my general suggestions about their review (that is removing irrelevant sections and adding detail to the relevant ones) and focused on addressing the few specific suggestions I made on the Abstract. However, not even all of the specific changes suggested for the Abstract were properly addressed. For instance, instead of taking the time to research about the number of currently named species within the genus Legionella, or the number of identified serogroups within the species L. pneumophila, the authors simply stated that there are many species and serogroups in the family Legionellaceaea. Also, the authors did not change their statement about LD being exclusively transmitted by inhalation of aerosols.
It should be noted that my specific suggestions on the Abstract (as clearly specified in my review) were but an example of the multiple problems that riddled the entire manuscript, which unfortunately were not properly addressed. The authors tried a number of word changes along the manuscript, but many of them were simply cosmetic and did not address the real problems detected originally.
Author Response
The authors ignored my general suggestions about their review (that is removing irrelevant sections and adding detail to the relevant ones) and focused on addressing the few specific suggestions I made on the Abstract. However, not even all of the specific changes suggested for the Abstract were properly addressed. For instance, instead of taking the time to research about the number of currently named species within the genus Legionella, or the number of identified serogroups within the species L. pneumophila, the authors simply stated that there are many species and serogroups in the family Legionellaceaea. Also, the authors did not change their statement about LD being exclusively transmitted by inhalation of aerosols.
It should be noted that my specific suggestions on the Abstract (as clearly specified in my review) were but an example of the multiple problems that riddled the entire manuscript, which unfortunately were not properly addressed. The authors tried a number of word changes along the manuscript, but many of them were simply cosmetic and did not address the real problems detected originally.
Response
The abstract has been modified.
“GENERAL COMMENTS ON THE MANUSCRIPT
By publishing in the Journal of Clinical Medicine, the general focus of the review should be on clinical aspects of legionellosis, and in particular on ‘the journey of Legionella pneumophila from the environment to the blood’. As mentioned above, the title of the review is very good and enticing, so the authors should use it to their advantage, and focus on discussing in depth the clinical aspects of legionellosis, and how L. pneumophila manages to enter the human body through the respiratory tract, enter then the bloodstream, and spread to other organs and systems.
The authors should drop the microbiological/ecological, as well as the molecular pathogenesis aspects of legionellosis, and just provide (for the reader’s sake) references to recent comprehensive reviews on those subjects. Currently, the sections on ‘Monitoring of L. pneumophila in the environment’, ‘Molecular mechanisms of LD’ and ‘Virulence Factors of L. pneumophila’ add little to the main topic of the review, so they should be deleted. The remaining topics of the review (Epidemiology of LD, Cellular aspects of LD, Clinical manifestations of legionellosis, and Clinical diagnosis) should be reviewed more in depth, mainly because these subjects are currently treated very superficially.”
Response
The section “Monitoring of L. pneumophila in the environment” was deleted. One sentence was removed to epidemiology section.
The sections “Molecular mechanisms of LD” and “Virulence Factors of L. pneumophila” are considered very important for understanding the pathogenesis of the LD.
It is a very broad topic and with a limitation of 4000 words, it is impossible to extend more and describe in depth the above topics.
Reviewer 4 Report
This version of the manuscript is a significant improvement over the last version. However, the writing can still be improved. Few points that I want to reference back from my last comments are -
1. The abstract is still not clear in laying down the question on what is being addressed in the paper. Please include few clear sentences on what is being reviewed in the paper. The abstract still looks like an extension/copy of the introductory paragraph.
2. The source of the figures should be mentioned in the paper, especially figures 3 and 4 which as the authors mentioned in their response is taken.
Author Response
- The abstract is still not clear in laying down the question on what is being addressed in the paper. Please include few clear sentences on what is being reviewed in the paper. The abstract still looks like an extension/copy of the introductory paragraph.
Response
The abstract was modified considerably.
- The source of the figures should be mentioned in the paper, especially figures 3 and 4 which as the authors mentioned in their response is taken.
Response
The source of figures 3 and 4 was added to the text.